# Chemical Migration from Beverage Packaging Materials—A Review

**Petra Schmid * and Frank Welle** 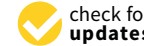

Fraunhofer Institute for Process Engineering and Packaging IVV, Giggenhauser Straße 35,
85354 Freising, Germany; frank.welle@ivv.fraunhofer.de

*   Correspondence: petra.schmid@ivv.fraunhofer.de

**Abstract:** The packaging of a beverage is an essential element for customer convenience and the preservation of beverage quality. On the other hand, chemical compounds present in the packaging materials, either intentionally added or non-intentionally, may be transferred to the food. With a huge variety of materials used in the production, beverage packaging requires safety assessments with respect to the migration of packaging compounds into the filled beverages. The present article deals with potential migrants from different materials for beverage packaging, including PET bottles, glass bottles, metal cans and cardboard multilayers. The list of migrants comprises monomers and additives, oligomers or degradation products. The article presents a review on scientific literature and summarizes European food regulatory requirements. The review shows no evidence of critical substances migrating from packaging into beverages. Testing the migration in real beverages during and at the end of the shelf life shows compliance with the specific migration limits. Accelerated testing using food simulants, however, shows higher migration in some cases, especially at high temperatures in ethanolic simulants. For some migrants, more realistic testing conditions should be applied in order to show compliance with their specific migration limits.

**Keywords:** migration testing; food simulants; food law compliance; PET bottles; glass bottles; beverage cans; cardboard packaging

## 1. Introduction

Beverages are one of the food categories with the broadest variety of packaging materials. Nearly all packaging materials are used for beverage packaging. The traditional packaging material for beverages is glass bottles. At the end of the last century, plastic bottles made from poly(ethylene terephthalate) (PET) were introduced into the beverage market. Compared to glass bottles, PET bottles are light-weight and unbreakable, which is an advantage in most cases. However, sensitive beverages like fruit juices and beer are still packed in glass bottles. On the other hand, less sensitive beverages like mineral water and soft drinks are increasingly filled in PET bottles over the last decades. Additionally, a traditional beverage packaging material is cardboard packaging. Developed and brought on the market in 1951, cardboard multilayer packaging, with or without aluminum films as oxygen barriers, is used today mainly for the packing of milk and juices, which are oxygen and light sensitive beverages. Metal cans can be also considered as traditional beverage packaging material and are mainly used today for beer and soft drinks. As a niche market, pouches with multi-layer film are also on the market for non-carbonated beverages. Within the last three decades, these traditional beverage packaging materials are losing market share towards PET bottles. However, not only the main packaging material for beverages should be taken into account. Minor packaging materials such as closures are important for the functionality and convenience of beverages packaging. Closures for bottles are composed

of polyolefins (polyethylene, polypropylene) or aluminum comprised coatings and polymer inlays. Cardboard packaging is also equipped with polyolefin closures.

Whereas cardboard and metal beverage packaging are only one-use packaging materials, bottles made of glass and PET may be refillable or non-refillable. In the case of glass, most of the bottles were refillable (multi-use) bottles. Regarding PET, the main market for PET are one-use bottles, but in some countries also refillable bottles have a significant market share. Due to the actual circular economy debate, PET bottles are increasingly made of post-consumer recyclates. PET non-refillable bottles with a recyclate content of up to 100% are on the market. Beverage packaging is also an interesting field in terms of active and intelligent packaging materials. Active oxygen scavengers, that are already used for oxygen sensitive beverages like juices or beer, can be mentioned as an example. A review on the recent trends in beverage packaging is given by Ramos [1]. However, active or intelligent packaging materials are only a niche market for beverages at the moment.

The main function of packaging is to protect the beverage from negative influences from the environment from the filling to the consumption. Beverage packaging prevents the loss of carbon dioxide in carbonated beverages. In the case of oxygen or light sensitive beverages, the packaging materials protect the beverage also from these negative influences. Another important function of beverage packaging is tamper-proofing. Beverage packaging also brings convenience to the customer.

In conclusion, several packaging materials were used today for the packing beverages. Each of these materials found their market due to its advantages (e.g., oxygen or light projection, light-weighting, inertness, etc.). On the other hand, consumer health protection must be guaranteed. Chemical migration from beverage packaging must be minimized. Thus, the overall migration limit as a measure for the inertness of a material and the specific migration limits for intentionally added additives or monomers must not be exceeded. This literature review focuses on the chemical migration from beverage packaging materials. From a legal point of view, the beverage packaging materials were evaluated according to European food law with their specific migration limits and testing conditions.

## 2. Chemical Migration from Beverage Packaging Materials

### 2.1. Food Law Compliance Testing of Beverage Packaging

2.1.1. Basic Food Regulatory Requirements in the European Union

Beverage packaging may be composed of a huge variety of materials including plastics, glass, metal and cardboard. In addition, printing inks or coatings may be used in the finishing of the final food contact article. Very often, a combination of different materials is used, for example a combination of metal and coating in beverage cans or an even more complex combination of plastic, aluminum, paper and ink in a multilayer fruit juice carton. When these materials come into contact with food, their compounds can be transferred from the packaging material into the food by a process called "migration". Migration can be characterized as a diffusion process following Fick's Law of diffusion. Basically, the transfer of packaging compounds into food may take place on two main pathways: direct transfer of components into the contacting food or transfer of compounds via the gas phase. In addition, set-off may be taken into consideration as an alternative route [2]. Gas phase transfer and set-off migration are of particular relevance for cardboard beverage packaging (see Section 2.5—Beverage Cardboard Packages).

In order to ensure the safety and quality of the packed food, binding food regulatory principles have been established in the European Union. Commission Regulation (EC) No 1935/2004 'on materials and articles intended to come into contact with food' [3] represents the European framework legislation for all materials and articles in food contact applications. Article 3 of this Regulation requires that food contact materials and articles do not transfer their constituents to food at levels harmful to human health. Furthermore, the packaging material must not change food composition or the organoleptic properties of the packed food in an unacceptable way. In addition to these general requirements applicable to all materials, specific materials or groups of materials may be subject to individual

measures. To date, specific regulations have been established for plastics, recycled plastics, ceramics, regenerated cellulose as well as active and intelligent materials and articles [4]. By contrast, European wide specific rules for widely used material groups such as paper and board, printing inks or coatings have not been adopted so far. These materials can be covered by national measures.

The most comprehensive specific European measure is Commission Regulation (EU) No 10/2011 [5] replacing the former Plastics Directive 2002/72/EC [6]. Regulation (EU) No 10/2011 is applicable to food contact materials and articles made of plastics (e.g., bottles made of polyethylene terephthalate, PET or closures made of high-density polyethylene, HDPE). Annex I of the Plastic Regulation establishes a positive list (so-called Union list) of substances that are authorized to be intentionally used in the manufacturing process of plastic materials and articles. The Union list comprises (a) monomers or other starting substances; (b) additives excluding colorants; (c) polymer production aids excluding solvents; (d) macromolecules obtained from microbial fermentation. Following a risk assessment performed by the European Food Safety Authority (EFSA) within the authorization process, specifications for individual substances or substance groups, quantitative restrictions or specific migration limits (SML) are set out in order to ensure the safety of the final material or article. The specific migration limit is defined as 'the maximum permitted amount of a given substance released from a material or article into food' [5] and is expressed in mg per kg food or beverages. Defined components are used in the manufacture of PET materials. The specific migration limits for some of those compounds are given in Table 1 (see Section 2.2—PET Bottles). For all other beverage packaging materials, the restricted compounds depend on the applied monomers and additives. The latter have to be evaluated on a case by case basis according to the Union list.

**Table 1.** Specific migration limits according to the EU Plastics Regulation for compounds used in the manufacture of PET bottles.

| Substance | Function | Specific Migration Limit (SML) * |
|---|---|---|
| Mono- and diethylene glycol (including the ester of stearic acid with ethylene glycol) | Monomer | 30 mg/kg |
| Terephthalic acid | Monomer | 7.5 mg/kg |
| Isophthalic acid | Monomer | 5 mg/kg |
| Antimony trioxide | Catalyst | 0.04 mg/kg (calculated as antimony) |
| 2-Aminobenzamide (anthranilamide) | Acetaldehyde scavenger | 0.05 mg/kg |

* SML: specific migration limit (expressed in mg substance per kg food) according to the EU Plastics Regulation [5].

In addition to substance related specific restrictions, the Plastics Regulation (EU) No 10/2011 lays down an overall migration limit (OML) of 10 milligrams for total non-volatile substances released per $dm^2$ of food contact surface into food simulants. The overall migration limit represents a generic limit for the inertness of the material and is determined by gravimetric measurements. It is the implementation of the Article 3 requirement of the Framework Regulation (EC) No 1935/2004 not to bring about an unacceptable change in the composition of the packed food.

It shall be ensured by the manufacturer that materials and articles for contact food will meet the food regulatory restrictions for the intended food contact application. For that purpose, migration experiments are carried out as an instrument for monitoring and controlling the transfer of components from the packaging material into the contacting food. The aim of migration testing is to measure and assess the (maximum) concentration of a migrating packaging component in food. Based on the results of migration experiments, compliance or non-compliance of the food contact material with the regulatory requirements is evaluated.

2.1.2. Migration Testing

A migration test is aimed at the determination and the assessment of the actual or the maximum concentration of a migrated packaging compound in food for a specific packaging application. The concentration in food is the result of the transfer of packaging components at the given conditions and is dependent on the properties of the packaging material and the filling good, the physico-chemical characteristics of the migrating substance and the storage conditions (time and temperature). From an analytical point of view, the analysis of migrating substances in real foodstuffs may be challenging due to the often complex food matrix, which may require a time and cost-intensive sample work-up. Therefore, migration experiments typically use food simulants instead of real food. Food simulants are test media mimicking real food but presenting a less complex matrix than food itself. Basically, the experiment comprises two elements: the migration contact step in which the material is brought in contact with the food simulant and the analytical determination of the concentration of the analyte in the food simulant as a second step, which is explained in more detail below. For materials and articles made of plastics, detailed rules form migration testing is laid down in Regulation (EU) No 10/2011. Conventional, overall and specific migration testing with food simulants is performed under standardized time and temperature conditions in order to obtain comparable results in the verification of compliance. The respective conditions and simulants shall be taken from Annex III (food simulant) and Annex V (time and temperature conditions) of the Regulation and shall be chosen based on the intended food contact application of the plastic material or article to be tested. The list of official food simulants according to Regulation (EU) No 10/2011 includes simulants for fatty, aqueous, acidic or dry food. Food simulants assigned to different beverage categories according to Regulation (EU) No 10/2011 are given in Table 2. Before the Plastics Regulation (EU) No 10/2011 entered into force in 2011, the rules on food simulants to be used for migration testing were laid down in Directive 85/572/EEC [7] and water was prescribed as the standard simulant for migration testing of packaging materials in contact with milk products. However, comparative studies on the migration of styrene from polystyrene milk packages in water and milk have shown that water does not exhibit the required physico-chemical properties for appropriately simulating the migration behavior of milk and milk products. Thus, migration testing using water resulted in an underestimation of the migration in real milk products [8]. This was also illustrated by the migration of the photo-initiator 2-isopropylthioxanthone (ITX) from laminate beverage carton packages with a printing ink applied on the non-food contact side. ITX can be transferred from the external printing ink to the food contact side of the laminate when stored on reels as a result of the so-called invisible set-off (see Section 2.5—Beverage Cardboard Packages). It was found that the actual packaged food-milk or cloudy juices showed higher ITX migration values than pure water as the fat content of milk and the fruit pulp in cloudy juices increase solubility of the substance [9]. As a consequence of these scientific findings, 50% ethanol was legally established as food simulant for milk products (and also for cloudy juices) in Regulation (EU) No 10/2011. The list of food simulants in Regulation (EU) No 10/2011 was generally shifted towards food simulants with higher contents of ethanol compared to the Directive 85/572/EEC (e.g., 10% ethanol instead of water, 20% ethanol instead of 15% ethanol). This was to increase solubility for lipophilic migrants and counteract the underestimation of migration into real foodstuffs with the previously assigned aqueous simulants [10]. For beverage packages intended for pure water, migration testing still can be performed using water. In this case, water represents the actual food. According to the provisions of Regulation (EU) No 10/2011, the results of a migration analysis using the actual food shall prevail over the results using a food simulant [5]. In the initial migration contact step, tests specimens of the material are brought in contact with the food simulant under the time and temperature combination as chosen from Annex V of Regulation (EU) No 10/2011. Beverage packaging is typically in contact with food for long term storage at room temperature or below. For specific migration testing and for storage times exceeding 30 days, the migration contact shall be performed at an elevated temperature for a maximum of 10 days at 60 °C based on the rules of Regulation (EU) No 10/2011. The test conditions of a maximum 10 days at a temperature of 60 °C should cover long-term

storage for more than 6 months at room temperature, including hot fill-conditions or heating up, e.g., to 70 °C for up to 2 h or to 100 °C for up to 15 min. For overall migration testing, migration contact shall be performed for 10 days at a temperature of 40 °C in order to cover long term storage at room temperature of below including hot fill-conditions according to Annex V (Table 3: standardized testing conditions for overall migration testing) of Regulation (EU) No 10/2011.

**Table 2.** Food simulants according to Regulation (EU) No 10/2011 assigned to different beverage categories.

| Beverage Category | Assigned Food Simulants * |
|---|---|
| Clear drinks, e.g.,: | Ethanol 20% (*v/v*), food simulant C |
| water, soft drinks, clear fruit or vegetable juices, tea, coffee | ** Acetic acid 3% (*w/v*), food simulant B |
| Cloudy drinks, e.g.,: | Ethanol 50% (*v/v*), food simulant D1 |
| juices, nectars and soft containing fruit pulp | ** Acetic acid 3% (*w/v*), food simulant B |
| Alcoholic beverages, alcoholic strength between 6%vol and 20%vol | Ethanol 20% (*v/v*), food simulant C |
| Alcoholic beverages, alcoholic strength above 20%vol and cream liquors | Ethanol 50% (*v/v*), food simulant D1 |
| Milk and milk-based drinks | Ethanol 50% (*v/v*), food simulant D1 |

* food simulants according to the EU Plastics Regulation [5]. ** migration testing in food simulant B can be omitted if the beverage has a pH value of more than 4.5.

**Table 3.** Rating scale according to DIN 10,955 used for the intensity of the difference in odor and taste compared to the reference.

| Intensity | Description |
|---|---|
| 0 | No perceptible change in smell or taste |
| 1 | Perceptible change in smell or taste |
| 2 | Slight change in smell or taste |
| 3 | Noticeable change in smell or taste |
| 4 | Very noticeable change in smell or taste |

The testing procedures for migration contact (e.g., migration contact by total immersion in the food simulant or by article filling) have been developed by the European Committee for Standardization (CEN). The methodologies for specific migration testing are laid down in the CEN standard EN 13130-1 [11], the contact procedures for overall migration testing can be found in the CEN standard EN 1186 series [12]. Following the prescribed migration contact, specific analytical methods are applied to determine the concentrations of specific migrants in the food simulant whereas a gravimetric analysis is used for the determination of the overall migration value. For the food simulants 3% acetic acid, 10% ethanol, 20% ethanol and 50% ethanol (corresponding to food simulants A, B, C and D1) according to Annex III of Regulation (EU) No 10/2011, the simulant is evaporated to dryness after the migration contact period and the residue is determined by weighing.

The analytical approach and the detection methods used for measuring the concentration of a specific migrant in the migration solutions is dependent on several parameters, including the volatility and polarity of the substance, the required detection and quantification limit or the used food simulant (e.g., aqueous or fatty). Consequently, a variety of instrumental techniques are applied ranging from gas chromatography (GC) to high performance liquid chromatography (HPLC) coupled to various detection systems (e.g., MS, FID, UV detection). As a general requirement, the analytical methods for the analysis of a migrating substance should be applicable and suitable for the intended purpose and comply with the performance criteria (such as selectivity for the analyte to be determined, repeatability, recovery, linearity of the calibration) as defined in Article 11 of Regulation (EC) No 882/2004 [13].

2.1.3. Non-Intentionally Added Substances

In addition to the substances, which are intentionally used in the manufacture of the polymer or during production of the beverage packaging material (e.g., monomers and additives), also substances that have not been added for a technical reason can be present in beverage packages. These non-intentionally added substances (NIAS) are, in most cases, impurities in the packaging

material like traces of solvents or degradation or reaction side-products of additives or the polymer itself [14]. It is current practice to evaluate the migration of NIAS into food with a specific migration limit of 10 µg/L. For the evaluation of migration of oligomers, a specific migration limit of 50 µg/L is typically applied [15]. In comparison to the specific migration limits of monomers and additives given in Table 1, a concentration of 10 or 50 µg/L in beverages is very low. As a consequence, the evaluation of NIAS is an important aspect for the food law compliance of beverages packaging materials.

### 2.1.4. Expression of Migration Test Results

In the positive list in Annex I of the Plastics Regulation (EU) No 10/2011, the specific migration limits for authorized substances are given in mg/kg food. Following the analytical determination of the concentration of a migrant in a food simulant, the obtained results need to be compared to these legislative limits in order to evaluate food regulatory compliance and establish consumer safety. For the migration experiment, a defined surface of the article to be tested is brought in contact with a defined quantity of the food/food simulant. The result of the migration test expressed in mg/kg needs to be corrected when the migration analysis has been performed applying a surface-to-volume ratio differing from the surface-to-volume ratio in the intended application. According to the provisions of Regulation (EU) No 10/2011, for some articles, the migration test results shall not be corrected for the surface-to-volume ratio under the actual or intended use, but by a standardized surface-to-volume ratio of 6 dm$^2$ per kg food (EU cube model). This is for example the case for beverage packages that are intended to be in contact with less than 500 mL.

The overall migration value is expressed as an area-related value regardless of the size or the filling volume of the article. The only exception is given for plastic materials and articles that are intended for infants and young children. In this case, the overall migration is linked to the filling volume of the packaging and a filling related overall migration limit of 60 mg/kg is set by the Plastics Regulation (EU) No 10/2011.

For a bottle-closure-system, both components have to be considered for a food regulatory compliance evaluation. In addition to the overall migration or specific migration of a bottle, the migration from the corresponding cap needs to be taken into account. Sealing devices such as caps or stoppers for beverage packaging are normally only in contact with food via a small food contact area (in the range of 3 to 7 cm$^2$ for typical bottle closures) compared to the contact area of the container (in the range of 3 to 8 dm$^2$ for typical bottles). A sealing device therefore contributes only to a limit extent to the total migration of substances from the closed article into the food. This aspect is respected in the regulatory provisions. In the particular case of caps or other sealing articles, the specific migration shall be expressed in mg/kg using the actual content of the container for which the closure is intended and applying the total food contact surface of sealing article and sealed container if the intended use of the article is known (Art. 17 (3) of Regulation (EU) No 10/2011). With regard to the overall migration, the results shall be expressed in mg/dm$^2$, applying the total contact surface (food contact area of the sealing article and sealed container), if the intended use is known (Art. 17 (4) of Regulation (EU) No 10/2011). If the intended use is of a sealing device is not known, the overall and specific migration results can be expressed in mg/article for both the overall and specific migration.

### 2.1.5. Sensory Compliance Evaluations

It is a key regulatory requirement that a packaging material shall not impact food quality. This is laid down in Article 3 of the Framework Regulation (EC) No 1935/2004. This article demands that the packaging material must not change the organoleptic properties (visual appearance, taste and smell) of the packed food in an unacceptable way. Acetaldehyde, a degradation product of the PET polymer, is a potential off-taste source for carbonated water filled and stored in PET bottles. The organoleptic threshold value for acetaldehyde is in the concentration range of 10 µg/L to 25 µg/L (see Section 2.2.3—Acetaldehyde). Trace amounts of acetaldehyde transferred from the PET bottle wall into water may exhibit an undesired, sweet off-taste of the water. Acetaldehyde is not stable in

non-carbonated water, so in most cases carbonated water is affected by this off-taste [16]. For juices, soft drinks or beer, acetaldehyde that has migrated from the bottle wall into the filling good is negligible as acetaldehyde is contained in these foodstuffs in even higher concentrations [17,18]. PET packaging may release also higher molecular weight aldehydes which might change the organolectic properties of the packaging material [19]. Additionally, in beverage bottle closures made of HDPE material, higher molecular weight aldehydes (such as octanal or nonanal) or carboxylic acids (e.g., propanoic acid, butyric acid) may be found as degradation products from the HDPE polymer or added masterbatches [20]. Aldehyde and other carbonyl compounds are also generated from polyolefin closures especially under sunlight exposure [21]. Such substances are also present as aroma compounds in food and odor-active even in low concentrations. Consequently, migration of aldehydes or carboxylic acid may have adverse effects on the organoleptic properties of the filled beverages which is opposed to the requirements of Article 3 of Regulation (EC) No 1935/2004.

The given examples illustrate that organoleptic tests are an essential aspect in terms of quality control of beverage packaging materials and are mandatory for food regulatory compliance evaluations. Sensory testing is commonly performed according to the Standard DIN 10955 [22]. The aim of these tests is to evaluate if a packaging material has any inherent odor or may transfer sensory active components to a test food at defined testing conditions. As test food, either the original filling good or simulating food (e.g., water as a simulant for aqueous food or milk as a simulant for aqueous low-fat food) is used. The sensory analyses for odor and taste transfer are conducted in comparison to a reference, which is represented by the corresponding test food that was not in contact with the packaging material. DIN 10955 requires that the analysis is performed by a panel of at least six selected panelists trained for perceiving, describing and rating a packaging related off-odor or off-taste. For panel qualification, regular exercises shall be performed. The intensity of the difference in odor and taste compared to the reference is rated according to a rating scale as given in DIN 10955 ranging from an intensity of zero for "no perceptible change in smell or taste" to an intensity of 4 for "very noticeable change in smell or taste" (Table 3). The obtained intensity value is then used as a basis for the food regulatory assessment with regard to the requirements of Article 3 of Regulation (EC) No 1935/2004. Typically used as threshold limit for sensory compliance of packaging materials is an intensity of 2.5.

## 2.2. PET Bottles

Poly(ethylene terephthalate) (PET) bottles are widely used for beverages. Knowledge about the migration of organic compounds from the PET bottle wall into contact media is of interest. A comprehensive literature review on the compliance of PET materials had been published by Welle [18]. Franz and Welle [23] published a systematic study on the migration of model substances into beverages (orange juice, apple juice, vitamins ACE juice, flavored water, cola) and food simulants (3% acetic acid, 10% ethanol, 50% ethanol and 95% ethanol). However, a food simulant may not exactly simulate the real migration into beverages. Therefore, typically a worse-case simulation behavior is the intention when using food simulants or when using diffusion modelling. The study investigates the migration kinetics of low molecular weight compounds with relatively high diffusion coefficients and, therefore, with high migration potential from the PET bottle wall into beverages. The results show, that food simulants like 3% acetic acid, 10% ethanol or isooctane do not swell the PET bottle wall material and result in migration levels similar to beverages. On the other hand, the simulant 95% ethanol show swelling effects and an over-estimative migration behavior compared to real beverages. The authors concluded that the most appropriate food simulant for PET packed foods with a sufficient but not too over-estimative worse-case character is 50% ethanol. In addition, the mass transport from PET is generally controlled by the very low diffusion in the polymer. An important consequence is that migration levels from PET food-contact materials are largely independent from the nature of the packed food, which on the other hand simplifies exposure estimations from PET bottles. Another systematic study on the migration from PET bottles was published by Gehring and Welle [24]. The authors investigate the migration into several food simulants (3% acetic acid, 10% ethanol, 20%

ethanol, 50% ethanol, 95% ethanol and isooctane) at temperatures of 40, 50 and 60 °C. The authors concluded that testing conditions of 10 days at 60 °C strongly over-estimated the migration under room temperature conditions. For example, the migration value of 2-aminobenzamide after 10 days at 60 °C corresponded to a storage time of 11.7 years at 23 °C, which is significantly higher than the shelf life of beverages in PET bottles and the long-term storage of more than 6 months as prescribed in Regulation (EU) No 10/2011. For larger molecules than 2-aminobenzamide, the corresponding storage times at room temperatures would even be longer. As a result, the migration at the end of shelf life would be over-estimated by far. Contact conditions of 10 days at 60 °C are too severe for PET bottles. The authors concluded that migrant- and polymer-specific diffusion parameters should be considered when designing accelerated migration tests.

### 2.2.1. Specific Migration of Monomers and Overall Migration

Störmer et al. [25] investigated the specific migration of the monomers monoethylene and diethylene glycol, as well as of terephthalic acid. In addition, the overall migration was tested on 34 individual PET bottles from the German market. The migration was determined using the food simulants 3% acetic acid, 10% ethanol and 95% ethanol at contact conditions of 10 days at 40 °C. In all cases overall migration was below of 0.5 mg/dm$^2$ and specific migration was close to or below the detection limits of the analytical methods. The authors concluded that due to the physical properties of the PET bottle material, the overall migration limit of 10 mg/dm$^2$ and the relevant specific migration limits for ethylene glycol, diethylene glycol and terephthalic acid cannot be exceeded. Only substances on or in the surface of the material may have an impact on the overall migration value. Therefore, a very rapid surface rinsing test, more rapid than the test conditions 24 h at 50 °C with 95% ethanol would be sufficient to show compliance with the overall migration limit. Ashby [26] also concluded that the overall migration on PET bottles was very low (below 0.6 mg/dm$^2$) for food simulants water, 3% acetic acid, 15% ethanol and 50% ethanol after 40 °C for 10 days.

A comprehensive study on the specific migration of terephthalic acid was conducted by Park et al. [27]. The authors analysed the migration of terephthalic acid, isophthalic acid and terephthalic acid dimethyl ester from 56 PET bottles and trays. The specific migration was determined into water, 4% acetic acid, 20% ethanol, and n-heptane at 60 °C for up to storage times of 30 days. As a result, the specific migration was found in all cases below of 0.1 μg/L. Ashby [26] investigated the specific migration of terephthalic acid and isophthalic acid into 3% acetic acid, 15% ethanol, 50% ethanol, vodka, and olive oil after storage for 10 days at 40 °C. As a result, the migration of terephthalic acid and isophthalic acid was below of the analytical detection limits of 10 μg/L and 50 μg/L, respectively. In swelling food or food simulants like vodka, 50% ethanol and olive oil, the concentrations of terephthalic acid are found to be higher in the range of 20 to 30 μg/L. The migration of isophthalic acid was still below the analytical detection limit, which might be due to the fact, that the concentration of isophthalic acid in PET is much lower than that of terephthalic acid. Gehring and Welle [24] found that the specific migration of ethylene glycol and diethylene glycol in 95% ethanol at 10 days at 40 °C and 10 days at 60 °C were below the detection limit of 200 and 150 μg/dm$^2$, respectively. In addition, terephthalic acid and isophthalic acid were not detectable in any of the investigated migration solutions at a detection limit of 26 μg/dm$^2$. As an overall conclusion from all studies, the specific migration of PET monomers like ethylene glycol, diethylglycol, terephthalic acid, isophthalic acid and terephthalic acid dimethylester is negligible under typical storage conditions of beverages.

### 2.2.2. Antimony

In principle, nearly every polymerization reaction needs a catalyst. Antimony trioxide (Sb$_2$O$_3$) or the reaction product with ethylene glycol is widely used as a polycondensation catalyst in the manufacturing of PET. The antimony trioxide has a high catalytic activity and has a low tendency to catalyze side reactions. In addition, antimony trioxide does not engender undesirable colors [28]. In principle, the polymerization catalyst remains in the PET polymer and is available for the migration

into contact media. Migration of antimony (Sb) from PET bottles into bottled mineral water was reported by Shotyk et al. in 2006 [29]. In a follow-up study, antimony concentrations were determined in 132 brands of bottled water from 28 countries [30]. As a result, leaching of antimony from PET bottles into mineral water shows a broad variability. In 14 brands of bottled water from Canada concentrations increased on average 19% during 6 months storage at room temperature. On the other hand, the antimony concentration of 48 water brands in 11 European bottled waters increased on average 90% under similar conditions. Two of the brands were at or above the maximum permitted antimony concentration for drinking water in Japan of 2 µg/L) [30], but far below the packaging related threshold limit in Japan of 50 µg/L [31]. Elevated concentrations of antimony in bottled waters are due mainly to the $Sb_2O_3$ used as the catalyst in the manufacture of polyethylene terephthalate [30,31]. Reimann et al. [32] mentioned that the median concentration of the waters sold in PET bottles is 21 times higher than for the same water sold in glass bottles. Although the leaching test demonstrates that antimony can also leach from glass bottles (the highest observed leaching value of antimony was from a glass bottle). A storage test demonstrates that antimony is migrating from the PET bottles. The authors found that the migration is almost independent from the pH value or the color of the PET bottles.

Welle and Franz [31] summarized the antimony levels found both in PET materials as well as in foods and food simulants. 67 PET samples from the European bottle market were investigated for their residual antimony concentrations. A mean concentration level of antimony of 224 ± 32 mg/kg was found in European PET bottles. The median was 220 mg/kg. In addition, the diffusion coefficients for antimony in PET bottle materials were experimentally determined at different temperature between 105 and 150 °C and activation energies of diffusion were determined. Using migration theory, the migration of antimony can be calculated if the antimony concentration in the bottle wall is known. On the other hand, for any given specific migration limit or maximum target concentration for antimony (or chemical compounds in general) in the bottled foodstuffs, the maximum allowable concentrations in the PET bottle wall can be calculated. Since a food simulant cannot exactly simulate the real migration into the foodstuff or beverages, a worse-case simulation behavior is the intention. The'results of the migration calculations were compared with literature data as well as international legal limits and guidelines values for drinking water and the migration limit set by food packaging legislation. The authors concluded that the migration of antimony from PET bottles cannot exceed the European-specific migration limit of 40 µg/L. Maximum migration levels caused by room temperature storage even after three years will never be essentially higher than 2.5 µg/L and in any case will be below the European limit of 5 µg/L for drinking water [33]. Keresztes et al. [34] reported antimony concentrations of 210–290 mg/kg in PET from Hungarian mineral water bottles, and significant leaching of antimony to the water depending on storage conditions and bottle volume. However, the increase of the antimony concentration was in the range of 0.7–0.8 µg/L after storage for one year and not exceeding a concentration of 1 µg/L after a three-year storage time. Even under extreme light and temperature storage conditions, the antimony concentrations were in the range of 2 µg/L. The results of Keresztes et al. are in good agreement with those of Welle and Franz [31]. In conclusion, the migration of antimony into beverages is far below the legal threshold limit in Europe (Table 1). In addition, the tolerable daily intake (TDI) for antimony is 6 µg per kg BW. [35], which results in an intake of 360 µg per day for an adult with 60 kg. Drinking of 3 L mineral water from PET bottles with a concentration of 2 µg/L will result in only 1.7% of the TDI for an adult. An infant with 5 kg body weight (BW) drinking 0.75 L mineral water from a PET bottle [36] results in only 5% of the TDI. The migration of antimony from PET bottles into beverages is therefore from a consumer protection point of view negligible.

### 2.2.3. Acetaldehyde

Acetaldehyde is a degradation product of the PET polymer and can be determined in every PET bottle. Acetaldehyde is generated at high temperatures in the melt phase of the PET during the manufacturing of the PET bottles. Typically, the concentrations of acetaldehyde in PET bottles are in

the range of 1 mg/kg (mineral water bottles) to 10 mg/kg (soft drink bottles). The lower acetaldehyde concentration found in mineral water bottles is due to the use of acetaldehyde scavenging agents (see Section 2.2.4—2-Aminobenzamide) as well as due to special PET polymer grades with a lower acetaldehyde forming potential. The specific migration limit of acetaldehyde is 6 mg/L (Table 1). With a typical PET bottle weight of 25 g, a concentration of acetaldehyde of 10 mg/kg and assuming total mass transfer into 1 L mineral water, the concentration of acetaldehyde in food can be calculated to be 250 µg/L. With a factor of 24, the migration is well below the specific migration limit for acetaldehyde according to EU legislation. Therefore, the conclusion can be drawn that the specific migration limit of acetaldehyde in food generally cannot be exceeded. On the other hand, the organoleptic threshold concentration of acetaldehyde in (mineral) water is low. It lies between 10 µg/L (retro-nasal) and 25 µg/L (ortho-nasal) [37]. Migration of acetaldehyde from the PET bottle wall into mineral water might result in an off-taste. In such cases, the PET bottle is not in compliance with sensory requirements of Article 3 of the European Framework Regulation 1935/2004 [3] (see Section 2.1.5—Sensory Compliance Evaluations). However, the migration of acetaldehyde and compliance with the Framework Regulation 1935/2004 is only relevant for mineral water. Soft drinks, fruit juices and beer have much higher concentrations of acetaldehyde as natural ingredient as the organoleptic threshold limits given above for acetaldehyde in water [17]. A literature review on the migration of acetaldehyde from PET bottles has been recently published [16]. Migration kinetics of acetaldehyde from PET bottles into mineral water as well as the diffusion coefficients derived thereof are published by Ewender et al. [38]. Diffusion coefficients of acetaldehyde in PET at several temperatures and the activation energies of diffusion in PET are also reported by Welle and Franz [39]. In conclusion, the specific migration limit set by Regulation (EU) No 10/2011 can never be exceeded due to the low concentrations in the PET bottle wall even if a total mass transfer of acetaldehyde is assumed. However, in the case of PET bottles for mineral water, the organoleptic threshold limit might be exceeded. The acetaldehyde concentration in PET bottles mineral water at the end of shelf life should be therefore monitored during routine controls.

### 2.2.4. 2-Aminobenzamide

Trace levels of acetaldehyde can migrate into natural mineral water during the shelf life and might influence the taste of the PET bottled water (see Section 2.2.3—Acetaldehyde). In order to reduce the concentration of acetaldehyde in the PET bottle wall, chemical scavengers are given into the PET melt during bottle manufacturing. 2-Aminobenzamide (anthranilamide) is such a chemical scavenger for acetaldehyde widely used in PET bottle manufacturing. Typically, a huge excess of 2-aminobenzamide is added into the PET melt and the acetaldehyde scavenging agent itself might migrate into mineral water. Franz et al. [40] determined the migration kinetics of 2-aminobenzamide into mineral water as well as into the food simulant 20% ethanol. The concentration of 2-aminobenzamide in mineral water after storage for 60 days at 40 °C was determined to 26.6 µg/L to 55.3 µg/L depending on the bottle size and 2-aminobenzamide concentration in the bottle wall. The specific migration value of 50 µg/L according to the requirements of the Regulation (EU) No 10/2011 [5] is exceeded in some cases. On the other hand, room temperature storage at the end of the shelf life results in significantly lower concentrations between 14.1 µg/L and 25.7 µg/L. Under these realistic storage conditions, the specific migration of 2-aminobenzamide was in all cases in compliance with the legal migration limit. Systematic migration studies into food simulants (3% acetic acid, 10% ethanol, 20% ethanol, 50% ethanol, 95% ethanol and isooctane) were published by Gehring and Welle [24,41]. The authors concluded, that migration testing for 10 days at 40 °C into non-swelling food simulants like 3% acetic acid, 10% ethanol and 20% ethanol approximately simulates the storage of 1-year shelf life at room temperature. Additionally, storage for 10 days at 60 °C with the non-swelling simulant isooctane was in good agreement with the migration values for room temperature till the end of the shelf life. However, when using actual legally recommended storage conditions (10 days at 60 °C, see Section 2.1.2—Migration Testing) into 3% acetic acid and ethanolic food simulants results in a strong over-estimation of the migration compared to the values at the end of shelf life. These accelerated testing conditions are

therefore not suitable for compliance evaluation of 2-aminobenzamide in PET bottles. In principle, the over-estimative character of accelerate migration tests into food simulants is desired. In most cases this over-estimative character of the simulant tests gave additional safety factors and in most cases, compliance with Regulation (EU) No 10/2011 can be still shown with accelerated migration tests in simulants (see Section 2.2.1—Specific Migration of Monomers and Overall Migration). However, in the special case of 2-aminobenzamide the compliance with Regulation (EU) No 10/2011 cannot be shown with accelerated tests in swelling food simulants. This is due to the fact, that 2-aminobenzamide is a small molecule with a corresponding high diffusion rate and that the specific migration limit is relatively low, which might be exceeded if the testing conditions are too severe. In conclusion, the acetaldehyde scavenging agent 2-aminobenzamide shows a high migration potential due to its high concentration in the PET bottle wall. The specific migration limit given by Regulation (EU) No 10/2011 might be exceeded if the actual legally recommended storage conditions of 10 days at 60 °C are applied. End of shelf life testing, however, results in migration values which are in agreement with the specific migration limit given in Regulation (EU) No 10/2011.

### 2.2.5. PET Oligomers

During PET polymerization and processing, oligomers are generated in the PET melt and thus not intentionally added to the polymer. Consequently, oligomers are often classified as NIAS (see Section 2.1.3—Non-Intentionally Added Substances) [42]. Oligomers with a molecular weight of up to 1000 g/mol are considered as relevant in terms of migration and risk assessment as it is conventionally assumed that larger molecules are not likely to be absorbed by the gastro-intestinal tract [43]. In scientific opinions of the European Food Safety Authority (EFSA) on two co-monomers for polyester type food-contact materials, a limit of 50 µg/L was defined for migration of total oligomers of less than 1000 g/mol [15,44]. The co-monomers and the migration limit for their oligomers were included in Regulation (EC) No 10/2011 with the 6th Amendment, Regulation (EU) No 2016/1416 [45]. The concentrations of PET oligomers in PET bottles are given in Hoppe et al. [46]. The highest concentrations are found for the PET cyclic trimer with 2922 mg/kg, followed by the cyclic tetramer and pentamer with 749 mg/kg and 303 mg/kg, respectively. Hoppe et al. investigated also the migration kinetics of PET oligomers into the food simulant 50% ethanol at 80 °C over a period of 15 h. Diffusion coefficients of five PET oligomers were calculated from the obtained data and compared with the predicted diffusion coefficients using the models of Welle [47] and Piringer [48]. The migration of cyclic oligomers into PET packed beverages were determined to be below the analytical detection limit of 0.05 mg/kg and 0.29 mg/kg in a study of Castle et al. [49]. As found by Hoppe et al. [46] the concentration of the cyclic trimer is significantly higher compared to the concentrations of the other oligomers such as dimers or tetramers. Consequently, the values presented above are primarily related to the cyclic trimer of PET. The diffusion coefficients of the cyclic trimer of PET at high temperatures were determined experimentally at 176 °C, 146 °C and 115 °C [50]. At lower temperature a determination of the diffusion coefficients might be not possible due to the very low migration of the oligomers. In conclusion, the migration of PET oligomers will come more and more into the focus of compliance evaluation. Oligomers are considered as NIAS with a relatively low migration limit of 50 µg/L and with high concentrations in the PET bottle wall, especially for the PET cyclic trimer. The migration limit of 50 µg/L is exceeded for high temperature applications.

### 2.3. Glass Bottles

Glass bottles are compared to PET bottles very inert packaging materials. This high inertness of the materials minimizes the migration of material components into food. However, glass bottles can release inorganic compounds into food. Misund et al. [51] reported multi-element (66 elements) concentrations in 56 European bottled mineral waters. Their study was not designed to detect contamination from bottle material, however these authors found clear indications of the release of Pb and Zr from glass bottles into bottled water. Reimann et al. [32] tested 294 samples of the same bottled water sold in glass

and PET bottles in the European Union. Within the investigated samples, 57 chemical elements were determined by inductively coupled plasma quadrupole mass spectrometry (ICP-QMS). It is important to note, that leaching of inorganic elements from the bottle materials occurs on top of their natural variation in mineral water. Therefore, the relative importance of leaching of inorganic elements from the bottle wall in relation to natural variation needs to be established. The concentrations of inorganic elements were compared between PET bottles and glass bottles. As a result, antimony has a 21× higher median value in bottled water when sold in PET bottles (0.33 µg/L vs. 0.016 µg/L). Glass contaminates the water with Ce (19× higher than in PET bottles), Pb (14×), Al (7×), Zr (7×), Ti, Th (5×), La (5×), Pr, Fe, Zn, Nd, Sn, Cr, Tb (2×), Er, Gd, Bi, Sm, Y, Lu, Yb, Tm, Nb and Cu (1.4×). The authors also investigated the influence of the color on the release of inorganic elements from the bottle wall. 136 bottles of the same water sold in green and clear glass bottles demonstrates an important influence of color. Water sold in green glass shows significantly higher concentrations in Cr (7.3×, 1.0 vs. 0.14 µg/L), Th (1.9×), La, Zr, Nd, Ce (1.6×), Pr, Nb, Ti, Fe (1.3×), Co (1.3×) and Er (1.1×) compared to water filled in clear glass bottles. A higher Cr content in green bottled water samples was also found by Marcinkowska et al. [52]. The diffusion process from glass should follow a different mechanism and is more likely due to corrosion of the glass [32]. For example, leaching of Pb to water stored in glass bottles should not take place due to diffusion coefficients being far too low. On the other hand, in a follow-up study Reimann et al. [53] investigated the leaching of elements from glass bottles into water at temperatures between 2 °C to 80 °C. As a result, certain elements (Ag, Al, As, B, Ba, Ca, Co, Cr, Cs, Cu, Fe, Ga, Ge, K, La, Li, Mg, Mo, Na, Ni, Pb, Rb, Sb, Se, Sn, Sr, Ti, U, V, W and Zr) leaching increases with storage temperature, while others (Be, Bi, Br, Cd, Ce, Dy, Er, Eu, Gd, Hf, Hg, Ho, I, Lu, Mn, Nb, Nd, Pr, Sc, Sm, Ta, Tb, Te, Th, Tl and Tm) appear unaffected by temperature. The authors concluded that storage conditions are important for bottled water quality and that storage of water in PET bottles at temperatures above 40 °C should be avoided. It is important to note, that none of the inorganic leachates approaches the maximum concentrations for drinking water [54] and mineral water [33].

## 2.4. Beverage Cans

Energy drinks, soft drinks, sparkling waters and beer are commonly filled in beverage cans. All beverage cans use an internal coating. Bisphenol A (chemical name: 4,4′-(propane-2,2-diyl)diphenol) is still present in the formulation of epoxy coatings of beverage cans, however also alternative coatings without bisphenol A are on the market. A review with concentrations of bisphenol A in food and consumer exposure is available in the scientific literature [55]. Following growing concerns on the endocrine-active properties of bisphenol A, migration of the substance is subject to restrictions. In the past decades, the assessment of bisphenol A has been in focus of toxicologists and the European legal bodies. In Europe, the use of bisphenol A as a monomer in the production of plastic materials and articles in contact with food, except infant feeding bottles, was authorized with a specific migration limit (SML) of 50 µg/kg food [56]. In France, the use of bisphenol A for any food packaging intended to come into direct contact with food is prohibited [57].

Several studies investigated the concentrations of bisphenol A (and other bisphenol derivatives) in canned beverages. Cirillo et al. investigated the contamination of bisphenol A, bisphenol B, bisphenol F, bisphenol A diglycidylether (BADGE) and bisphenol F diglycidylether (BFDGE) in 40 canned beers from the Italian market [58]. The results showed that only 14 samples were contaminated at concentrations ranging from 0.5 to 2.5 µg/L by at least bisphenol A, bisphenol F and BADGE. Bisphenol B and BDGE were not detected at a detection limit of 0.5 µg/L. The authors concluded that canned beers from the Italian market should represent neither a relevant source of intake of bisphenols nor a risk for consumer's health. Tzatzarakis et al. investigated the bisphenol A concentrations in beverages from the Greek market [59]. In 43.8% of the soft drinks' bisphenol A was detected. The mean concentration of bisphenol A in soft drinks was 2.30 ± 0.18 µg/L. Regueiro and Wenzl [60] analysed several canned and non-canned beverages. Only bisphenol A and three bisphenol F isomers were detected in any of the samples. Bisphenol A concentration ranged from <LOD to 1.26 ± 0.09 µg/L,

whereas bisphenol F varied from <LOD to 1.00 ± 0.08 µg/L. Fasano et al. [61] investigated the bisphenol A concentrations in carbonated, non-carbonated and milk-based beverages in a selection of brands that are mostly consumed by Italian children. Bisphenol A was found in 57% of carbonated beverages, in 50% of non-carbonated and in all investigated milk-based beverages. The median concentrations were 1.24 µg/L in carbonated beverages, 0.80 µg/L non-carbonated beverages and 3.60 µg/L milk-based beverages. The authors concluded, that the bisphenol A daily intake from sugary drink consumption in children ranged from 0.008 to 1.765 µg per kg BW per day. The median exposure values for the worst-cases was 0.47% respectively of the EFSA t-TDI for bisphenol A of 4 µg per kg BW per day for infants, and 10.59% and 35.30% of the t-TDI when the maximum levels were considered. Geens et al. [62] determined the concentrations of bisphenol A in 45 canned beverages from the Belgian market. The investigated beverages had an average concentration of bisphenol A of 1.0 µg/L, while bisphenol A was not detected in non-canned beverages at a detection limit of 0.02 µg/L. Cao et al. [63] investigates 72 canned soft drink products for bisphenol A. The concentration of bisphenol A was determined to 0.032 to 4.5 µg/L. About 75% of the products had bisphenol A levels of <0.5 µg/L, and 85% of the products had bisphenol A levels of <1 µg/L. The authors concluded, that the exposure to bisphenol A through consumption of canned soft drink products is low. In another study of Coa et al. [64] investigated 22 soft drink samples and 16 beer samples in both cans and plastic and/or glass bottles regarding their bisphenol A concentrations. As a result, bisphenol A was not detected in all soft drink samples in plastic or glass bottles except for one product with a very low bisphenol A concentration of 0.018 µg/L. In soft drinks packed in in cans the concentrations of bisphenol A was detected to 0.019 to 0.21 µg/L. Bisphenol A was detected in only one of the seven beer products in glass bottles (0.054 µg/L) but was detected in all beer samples in cans at concentrations of 0.081 to 0.54 µg/L. The results indicate that migration from can coatings is likely the source of BPA in canned beverages. Stärker and Welle 2019 [65] investigated the migration of bisphenol A from can coating into beverages and the food simulant 20% ethanol. The European official testing conditions with 20% ethanol for 10 days at 40 °C and 60 °C were compared to migration into beverages. As a result, migration of bisphenol A from bisphenol A-containing coatings was determined to be considerably higher when tested at a temperature of 60 °C in comparison to the tests performed at 40 °C. On the other hand, migration into energy drinks and cola, from the same coatings at the end of shelf life when stored at room temperature, was found to be below the detection limit in either case. Based on spiking tests, bisphenol A is stable in the investigated food matrix. The authors concluded that the accelerated migration tests using 20% ethanol as food simulant at the test conditions 10 days at 40 °C and 60 °C significantly over-estimates the actual migration into beverages at the end of shelf life. This over-estimation can be attributed to the swelling of the epoxy can coating caused by the ethanolic food simulant. As an overall conclusion from the data of the above-mentioned studies, the concentration of bisphenol A determined in beverages is in the lower µg/L range, which results in <1% of the t-TDI for bisphenol A of 4 µg per kg BW per day set by the EFSA.

### 2.5. Beverage Cardboard Packages

Beverage cartons are mainly used for filling and storage of juices, milk or cocoa at ambient or refrigerated temperature. These paper-based packaging products represent a more complex form of food packaging in terms of material composition as they normally contain several layers of the following materials: cardboard, aluminum, polymers, adhesives, printing inks applied on the non-food contact side. The complexity in composition adds to the number of potential migrants that can be transferred into the packed beverages. In the case of a beverage carton structure, the direct migration of compounds from the food contacting plastic layer is of relevance, and in addition, the chemical diffusion of compounds from the outside layers to the food contact layer has to be taken into consideration as well as the transfer of components via the gas phase. The latter process is of relevance for substance with a vapor pressure high enough to enter the surrounding air space. Gas phase transfer may be linked with an odor impact on the packed food [2]. The use of barrier materials (e.g., aluminum foils

or a barrier polymer such as ethylene vinyl alcohol copolymer, EVOH) can significantly reduce or even prevent migration of substances from the outside layer (e.g., the printing ink) to food via the diffusion process. However, even in presence of a barrier layer, constituents of the printing ink can be transferred to the unprinted food contact side by so-called invisible set-off which may occur when beverage carton sheets are stored in rolls during storage or transportation. In 2005, this set-off phenomenon related to the printing ink chemical isopropyl thioxanthone (ITX), resulting in a recall of baby milk filled in cartons due to safety concerns when public agencies in Germany and Italy detected residues of this photoinitiator in samples of the milk [66]. In view of the reports on the presence of this substance in food and the public concern, potential health risks associated with ITX were evaluated by the European Food Safety Authority (EFSA). EFSA concluded that performed toxicological studies did not indicate a genotoxic potential for the substance and that ITX did not give cause for health concerns at the detected concentrations [67].

Set-off cannot be ruled out per se for any printed material that is stored on rollers. According to the provisions of the European Regulation (EC) No 2023/2006, which lays down rules on good manufacturing practice for materials and articles intended to come into contact with food, printing inks applied to the non-food contact side shall be formulated and/or applied in such a manner that substances from the printed surface are not transferred to the food-contact side through the substrate or by set-off in concentrations that lead to levels of the substance in the food which are not in line with the requirements of Article 3 of Regulation (EC) No 1935/2004 [68]. The absence of harmonized European legislation for printing inks, but also paperboard or adhesives used in beverage carton packages poses a challenge to safety assessments and compliance evaluations according to the requirements of Article 3 of Regulation (EC) No 1935/2004. These materials may be subject to national rules. The most comprehensive legislation covering printing inks is established in Switzerland with Ordinance SR 817.023.21 [69]. This regulation includes a positive list of printing ink components that are allowed for the use in food contact applications. In addition, sectorial guidelines for non-specifically regulated packaging materials are published by industry, e.g., the 'EuPIA Guideline on Printing Inks' of the European Printing Ink Association [70]. The guideline offers recommendations as to how to formulate inks which will comply with the Framework Regulation (EC) No 1935/2004.

## 3. Conclusions

The food law compliance of beverage packaging materials—as any other packaging material—has to be ensured before entering the market. In the European Union, general food regulatory requirements with regard to safety and sensory aspects are laid down in the Framework Regulation (EC) No 1935/2004 which is applicable to beverage packaging composed of any material. For packaging made of plastics, specific requirements are set out in the European Plastics Regulation (EU) No 10/2011. This regulation also comprises detailed rules for the determination of the overall and the specific migration, including tables listing food simulants and time and temperature combinations that shall be applied for analytical migration analyses.

PET is a low diffusive polymer, which limits the migration of chemical constituents from PET bottles into beverages. The specific migration limits of the monomers as well as the overall migration limit cannot be exceeded. Additionally, the polymerization catalyst antimony cannot exceed the specific migration limit under normal storage conditions till the end of shelf life. This might be one of the reasons, why the use of PET bottles has strongly increased during the last two decades. Regarding mineral water packaging, the migration of acetaldehyde as well as 2-aminobenzamde should be controlled during routine testing. Acetaldehyde might change the organoleptic properties of the PET packed water. 2-aminobenzamide has a low specific migration limit which might be exceeded, especially when high temperatures are applied during accelerated testing conditions. In the case of 2-aminobenzamide, the regulated test conditions in most cases significantly over-estimate the actual migration into the packed beverages at the end of shelf life. Therefore, more realistic storage conditions should be applied for routine testing for 2-aminobenzamide. Glass bottles release inorganic compounds

into the beverages. The release is most probably more a corrosion process than a diffusion process. The release of inorganic compounds is, however, below any threshold concentrations given by drinking water regulations. Beverage can coatings still might release bisphenol A as the most critical compound. The concentration in beverages at the end of shelf life is below the specific migration limit. Accelerated testing with high ethanolic food simulants, however, result in some cases in higher migration results due to swelling of the coating during testing, especially at high temperatures. Beverage cartons represent a form of multilayer packaging often printed in different colors on the non-food contact side. It has to be taken into consideration that printing ink compounds can be transferred from the printed outside to the non-printed food contact side by means of invisible set-off. Furthermore, migration through the packaging material may take place unless barrier layers, e.g., aluminum foils, have been applied in the multilayer structure.

In conclusion, the current literature review of chemical migration from beverage packaging materials has not shown any critical compounds. Testing the migration in real beverages during and at the end of the shelf life shows compliance with the specific migration limits. Accelerated testing using food simulants at high temperatures results in significant higher migration and in some cases the threshold limits are exceeded. However, this is most probably due to unsuitable testing conditions and swelling of the packaging materials. In such cases, more realistic testing conditions should be applied in order to show compliance with their specific migration limits.

**Funding:** This research received no external funding.

**Conflicts of Interest:** The authors declare no conflicts of interest regarding the publication of this paper.

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
