# Peer review of "Chemical Migration from Beverage Packaging Materials—A Review"

_beverages, doi:10.3390/beverages6020037_

Round 1

Reviewer 1 Report

Overall, this is well written manuscript. The content and knowledge are quite deep enough to give reader useful information. The English proficiency is generally good. personally the manuscript can be accepted in its original form.

Author Response

Overall, this is well written manuscript. The content and knowledge are quite deep enough to give reader useful information. The English proficiency is generally good. personally the manuscript can be accepted in its original form.

Response to reviewer comment: Thanks

Reviewer 2 Report

This review discussed the different migrant compounds (comprising monomers, additives, oligomers, or degradation products) on packaging materials of beverages. This article also discusses the European food regulation regarding the packaging materials. Overall, the writing of this paper needs to improve as some sections have information with no logical connections, and they are difficult to understand. Moreover, I understand that this article uses the EU regulation for citing some of the methods, analysis, and results. However, doing this diminishes this paper's originality. For instance, most of the information that is associated with this paper can be found in the EU regulation about packaging materials. I suggest that this paper shits its focus on the scientific advances in measuring migration rates and the uses of novel equipment for rapid detections of migrants in packaging materials. This can add somewhat a bit more originality in this paper. More specific comments can be found below:

Formatting comment:

There are some minor formatting issues in this review that the authors need to address in the next version. For instance, the affiliation should have the email of the first author. The abstract and keywords titles should not be numbered. The division line between the keywords and the introduction is missing. These and other details should be carefully revised in the next draft. Besides, some sections also have references to chapters, but this review is not structured in chapters (maybe from a previous document or the EU regulation).

Abstract:

The abstract should be concise, but this might be too short to summarize the key points of this review. I suggest adding a couple of the main findings in regard to the migration of materials in the packaging of beverages.

3 Introduction:

Lines 21-39: The description of the different materials used in beverage packaging is not cohesive. That is, it seems that all the sentences were put together without a connector that can give fluidity to the paragraph. I suggest rewriting the first paragraph.

Lines 37-39: This sentence has to be expanded. This is the most important sentence of the introduction, as it is describing the whole structure of the review. It should contain more details about the specific sub-topics that will be covered.

4 Chemical Migration from Beverage Packaging Materials:

Lines 93-94: A better explanation of how the migration works is needed in this section. It is important to clarify how food stimulants are used as well as the test media, and how the different tests use the standardized time and temperature conditions for measuring migration rates.

Table 1: The citations/reference where these values were obtained must be stated (in the footnote of the table).

Table 2: Similar to Table 1, the citations/reference where this information was obtained must be stated (in the footnote of the table).

Lines 208-224: Something important to mention in the sensory tests is the description of the participants/panellists used for the evaluation. The regulation should indicate the adequate number of participants for this type of analysis and whether these participants are trained or untrained. A screening method of participants is recommended since some participants may have some sensory deficiencies such as ageusia or anosmia.

Lines 353-358: There is some discussion about how acetaldehyde can affect the sensory perception of some beverages. However, the authors only mentioned the sensory effects of acetaldehyde, and the effects of other compounds are not mentioned in the packing materials. This can be important for the discussion of the different off-tastes that the different materials have.

Lines 427-555: A table comparing the differences and similarities of the migration elements and processes of plastic, glass, cans, and cardboard can be included in this review as this can be useful for the overall understanding of those materials.

There is a lot of information for plastic; however, the discussion about glass, cans, and cardboard is very limited.

Author Response

This review discussed the different migrant compounds (comprising monomers, additives, oligomers, or degradation products) on packaging materials of beverages. This article also discusses the European food regulation regarding the packaging materials. Overall, the writing of this paper needs to improve as some sections have information with no logical connections, and they are difficult to understand. Moreover, I understand that this article uses the EU regulation for citing some of the methods, analysis, and results. However, doing this diminishes this paper's originality. For instance, most of the information that is associated with this paper can be found in the EU regulation about packaging materials. I suggest that this paper shits its focus on the scientific advances in measuring migration rates and the uses of novel equipment for rapid detections of migrants in packaging materials. This can add somewhat a bit more originality in this paper.

Response to reviewer comment: Our paper is aiming at providing a comprehensive overview on existing EU food law and important aspects of chemical migration from beverage packaging in the form of a review article. The topic of chemical migration from beverage packaging was agreed with the author, before we start writing this review article. Scientific advances or novel equipment could be subject to a separate paper but are not intended to be presented in this paper.

More specific comments can be found below:

Formatting comment:

There are some minor formatting issues in this review that the authors need to address in the next version. For instance, the affiliation should have the email of the first author. The abstract and keywords titles should not be numbered. The division line between the keywords and the introduction is missing. These and other details should be carefully revised in the next draft. Besides, some sections also have references to chapters, but this review is not structured in chapters (maybe from a previous document or the EU regulation).

Response to reviewer comment: We do not completely understand this point. We tried to consider the points. The following chapter structure is presented in this paper:

Abstract

Keywords

  1. Introduction
  2. Chemical Migration from Beverage Packaging

4.1 Food Law Compliance Testing of Beverage Packaging

4.2 PET bottles

4.3 Glass bottles

4.4 Beverage Cans

4.5 Beverage Cardboard Packages

  1. Conclusion
  2. Conflicts of Interest
  3. References

Abstract:

The abstract should be concise, but this might be too short to summarize the key points of this review. I suggest adding a couple of the main findings in regard to the migration of materials in the packaging of beverages.

Response to reviewer comment: We added some sentences with the major findings and conclusions.

3 Introduction:

Lines 21-39: The description of the different materials used in beverage packaging is not cohesive. That is, it seems that all the sentences were put together without a connector that can give fluidity to the paragraph. I suggest rewriting the first paragraph.

Lines 37-39: This sentence has to be expanded. This is the most important sentence of the introduction, as it is describing the whole structure of the review. It should contain more details about the specific sub-topics that will be covered.

Response to reviewer comment: We rewrote the Chapter introduction.

4 Chemical Migration from Beverage Packaging Materials:

Lines 93-94: A better explanation of how the migration works is needed in this section. It is important to clarify how food stimulants are used as well as the test media, and how the different tests use the standardized time and temperature conditions for measuring migration rates.

Response to reviewer comment: Amendments were made to the introduction of the “migration testing”-section in order to improve understanding.

Table 1: The citations/reference where these values were obtained must be stated (in the footnote of the table).

Response to reviewer comment: Reference to the EU Plastics Regulation for the given specific migration limits was added to the footnote.

Table 2: Similar to Table 1, the citations/reference where this information was obtained must be stated (in the footnote of the table).

Response to reviewer comment: Reference to the EU Plastics Regulation for the choice of food simulants has was added to the footnote.

Lines 208-224: Something important to mention in the sensory tests is the description of the participants/panellists used for the evaluation. The regulation should indicate the adequate number of participants for this type of analysis and whether these participants are trained or untrained. A screening method of participants is recommended since some participants may have some sensory deficiencies such as ageusia or anosmia.

Response to reviewer comment: The panel-related requirements according to DIN 10955:2004-06 were added with the following lines: “DIN 10955 requires that the analysis is performed by a panel of at least six selected panelists trained for perceiving, describing and rating a packaging related off-odour or off-taste. For panel qualification, regular exercises shall be performed.”

Lines 353-358: There is some discussion about how acetaldehyde can affect the sensory perception of some beverages. However, the authors only mentioned the sensory effects of acetaldehyde, and the effects of other compounds are not mentioned in the packing materials. This can be important for the discussion of the different off-tastes that the different materials have.

Response to reviewer comment: A detailed discussion on the specific topic of sensory properties of packaging components is not intended to be presented in this review paper. This would be an own review. However we added additional citations on the migration of aldehydes, which generates also off-flavours.

Lines 427-555: A table comparing the differences and similarities of the migration elements and processes of plastic, glass, cans, and cardboard can be included in this review as this can be useful for the overall understanding of those materials.

There is a lot of information for plastic; however, the discussion about glass, cans, and cardboard is very limited.

Response to reviewer comment: Well, this point reflects on the one hand the beverage packaging market. Plastics, especially PET, is the main packaging material for beverages. On the other hand, much more publications are published on the chemical migration from plastics compared to glass, cans and cardboard. Regarding cans, lots of publications can be found on cans and can coatings in general. However, publication on beverage cans are limited. Due to the fact, that only beverage packagings are in the focus of this review article, this non-balanced material selection is reflecting the scientific literature. Therefore no significant changes made in the manuscript according to this point.

Reviewer 3 Report

The manuscript deals with materials that are intended to be in contact with beverages and reviews different relevant aspects related with the chemical migration including legislation, migration test and relevant migrants (acetaldehyde, oligomers, etc.) as well as used food contact articles (e.g. PET, cardboard). The manuscript is interesting, well written as will be useful for scientist. There are few minor changes that the authors must address in order to publish the review:

Introduction is short, it should be extended.

Line 36- Specific and overall migration limit. Please include.

Line 44-Printing ink is not a material.

Lines 46-49- Substances may also migrate through gas phase. Since this review deals with printing inks, I think migration via gas phase should also be mentioned. It would make the review more robust.

Table 1 is confusing; a catalyst is an additive.

Lines 91-162- Migration Testing- What about for articles intended for repeated use?

Lines 93-113- If you talk about why 50% EtOH was introduced, the authors should also discuss why water was is no longer a food simulant. Also, why the other ‘’new’’ food simulants were introduced (10% EtOH, etc.)?

Line 162-190- ‘’Expression of Migration Tests Results’’- This part is a little confusing. Needs to be rewritten.

Line 194- Color is also an organoleptic property.

Lines 231-232-‘’ However, … cannot...‘’this may mislead the readers. Change by ‘’may not’’’ or’’ could not’’

Lines 389-393- This statement would need to be discussed in depth. If that is case, what do a food manufacture do? They follow your statement or the EU regulation? Have other authors observed and stated the same? Please include references. Shouldn’t be better to perform a migration test that overestimate migration? If that overestimation is significantly high, wouldn’t be better to divide by a reduction factor? What do the authors propose? I know this is controversial, that is why this needs to be discussed more in depth by the authors and be supported with references.

Lines 424-426-Yes, there are, please review literature again.

Line 427-459- ‘’Glass contaminates’’- The appropriate term is’’ Glass leaches or may leach’’, particularly if they comply with the stablished levels as the authors stated. In addition, not all glass leaches these inorganic compounds.

Lines 460-518- In this paragraph the authors should also include and discuss about the non-BPA based coatings, which are also widely used nowadays. I think it would be good to mention coatings regulation status as the authors did in the other paragraphs related to other materials.

Line 513-516- Here the authors should also include and discuss other published scientific papers where other authors have observed the contrary. There are studies where it is shown that migration at similar temperatures and using similar simulants migration is underestimated when compared with real food. Here is one of the most recent examples: Packaging Technology & Science (2020), 33(2), 75-82.

Line 580-581- This conclusion cannot be drawn. As stated in the previous paragraph, there are other studies that do not support this conclusion. Please modify accordingly.

After authors address and edit the review according to the comments presented above, the manuscript can be published.

Author Response

The manuscript deals with materials that are intended to be in contact with beverages and reviews different relevant aspects related with the chemical migration including legislation, migration test and relevant migrants (acetaldehyde, oligomers, etc.) as well as used food contact articles (e.g. PET, cardboard). The manuscript is interesting, well written as will be useful for scientist. There are few minor changes that the authors must address in order to publish the review:

Introduction is short, it should be extended.

Response to reviewer comment: We rewrote the Chapter Introduction.

Line 36- Specific and overall migration limit. Please include.

Response to reviewer comment: The overall migration limit was included as follows: “Chemical migration from beverage packaging must be minimized. Thus, the overall migration limit as a measure for the inertness of a material and the specific migration limits for intentionally added additives or monomers must not be exceeded.”

Line 44-Printing ink is not a material.

Response to reviewer comment: The sentences were improved as follows: “Beverage packaging may be composed of a huge variety of materials including plastics, glass, metal and, cardboard. In addition, printing inks or coatings may be used in the finishing of the final food contact article.”

Lines 46-49- Substances may also migrate through gas phase. Since this review deals with printing inks, I think migration via gas phase should also be mentioned. It would make the review more robust.

Response to reviewer comment: The section was expanded for a description of possible migration routes (including gas phase transfer). In addition, the following expansions were made in section 4.5 - Beverage Cardboard Packaging: “In case of a beverage carton structure, direct migration of compounds from the food contacting plastic layer is of relevance but in addition chemical diffusion of compounds from the outside layers to the food contact layer has to be taken into consideration as well as the transfer of components via the gas phase. The latter process is of relevance for substance with a vapour pressure high enough to enter the surrounding air space. Gas phase transfer may be linked with an odour impact on the packed food [2].”

Table 1 is confusing; a catalyst is an additive.

Response to reviewer comment: The function of anthranilamide was characterized (“acetaldehyde scavenger”) for clarification.

Lines 91-162- Migration Testing- What about for articles intended for repeated use?

Response to reviewer comment: The beverage packaging products in the center of this review (possibly with the exception of glass) are mainly no repeated use products (repeated contact with food by consumer use). Therefore, the repeated use testing procedure was not included in the “migration testing”-section. No changes made in the manuscript.

Lines 93-113- If you talk about why 50% EtOH was introduced, the authors should also discuss why water was is no longer a food simulant. Also, why the other ‘’new’’ food simulants were introduced (10% EtOH, etc.)?

Response to reviewer comment: The following amendment was introduced: “The list of food simulants in Regulation (EU) No 10/2011 was generally shifted towards food simulants with higher contents of ethanol compared to the Directive 85/572/EEC (10 % ethanol instead of water, 20 % ethanol instead of 15 % ethanol). This was to increase solubility for lipophilic migrants and counteract the underestimation of migration into real foodstuffs with the previously assigned aqueous simulants [9]. For beverage packages intended for pure water, migration testing can be performed using water. In this case, water represents the actual food. According to the provisions of Regulation (EU) No 10/2011, the results of a migration analysis using the actual food shall prevail over the results using a food simulant [4].”

Line 162-190- ‘’Expression of Migration Tests Results’’- This part is a little confusing. Needs to be rewritten.

Response to reviewer comment: The introductory paragraph of this section was rewritten to improve understanding.

Line 194- Color is also an organoleptic property.

Response to reviewer comment: “Visual appearance” was added.

Lines 231-232-‘’ However, … cannot...‘’this may mislead the readers. Change by ‘’may not’’’ or’’ could not’’

Response to reviewer comment: The sentence was changed as follows: “However, a food simulant may not exactly simulate the real migration into beverages.”

Lines 389-393- This statement would need to be discussed in depth. If that is case, what do a food manufacture do? They follow your statement or the EU regulation? Have other authors observed and stated the same? Please include references. Shouldn’t be better to perform a migration test that overestimate migration? If that overestimation is significantly high, wouldn’t be better to divide by a reduction factor? What do the authors propose? I know this is controversial, that is why this needs to be discussed more in depth by the authors and be supported with references.

Response to reviewer comment: The statement is given by the authors. Due to the fact, that this manuscript is a review, we adopted most of the findings and conclusions from the cited paper. In this special case, we tried to make it clearer, that this is not a general statement but a special case for 2-aminobenzamide.

Lines 424-426-Yes, there are, please review literature again.

Response to reviewer comment: We deleted the sentence. The literature is already cited in this chapter.

Line 427-459- ‘’Glass contaminates’’- The appropriate term is’’ Glass leaches or may leach’’, particularly if they comply with the stablished levels as the authors stated. In addition, not all glass leaches these inorganic compounds.

Response to reviewer comment: As the manuscript is a literature review, we used the wording of the authors. However, we agree, the using leaching is more neutral. Where possible we changed this paragraph accordingly.

Lines 460-518- In this paragraph the authors should also include and discuss about the non-BPA based coatings, which are also widely used nowadays. I think it would be good to mention coatings regulation status as the authors did in the other paragraphs related to other materials.

Response to reviewer comment: We cited the relevant publications on the chemical migration into food. Regarding beverage cans, most of these publications are dealing with bisphenol A and other bisphenols.

Line 513-516- Here the authors should also include and discuss other published scientific papers where other authors have observed the contrary. There are studies where it is shown that migration at similar temperatures and using similar simulants migration is underestimated when compared with real food. Here is one of the most recent examples: Packaging Technology & Science (2020), 33(2), 75-82.

Response to reviewer comment: This is a review and we adopted the conclusions of the authors. Regarding Packaging Technology & Science (2020), 33(2), 75-82. We read this paper and we do not agree with the statement, that the "migration" in simulants is under-estimative. Indeed, the "concentration" of bisphenol A is higher in real food, compared to simulants. However, this is an effect of the sterilisation process. In food simulants, bisphenol A is not regenerated during heating. This is also found in the paper of Staerker and Welle on beverage cans. In simulants, especially in ethanol simulants, a total extraction of the coating is observed. This is indicated in the similar migration results in all simulants independent of the storage conditions. In the meat balls, obviously a regeneration of bisphenol A occurs which increase the migration potential of bisphenol A. Due to the fact, that beverage cans are not sterilized we do not consider this point in our review.

Line 580-581- This conclusion cannot be drawn. As stated in the previous paragraph, there are other studies that do not support this conclusion. Please modify accordingly.

Response to reviewer comment: Se comment above. We tried to make clear in the manuscript, that this is a special case for simulant testing of bisphenol A and 2-aminobenamide and not a general statement.

After authors address and edit the review according to the comments presented above, the manuscript can be published.

Reviewer 4 Report

The manuscript entitled "Chemical Migration from Beverage Packaging Materials – a Review" is a reviw paper on migration phenomena, focusing on actual regulation and specific examples for beverage packaging materials. The manuscript is well written and the references are quite representative. The main concern is related to the missing information on "risk analysis" for NIAS compound. It is a critical aspect for producers of packaging materials and the authors did not discuss it al all. I woould suggest to include also this aspect to improve the quality of the review and to get the manuscipt useful for packaging material company. 

Author Response

The manuscript entitled "Chemical Migration from Beverage Packaging Materials – a Review" is a reviw paper on migration phenomena, focusing on actual regulation and specific examples for beverage packaging materials. The manuscript is well written and the references are quite representative. The main concern is related to the missing information on "risk analysis" for NIAS compound. It is a critical aspect for producers of packaging materials and the authors did not discuss it al all. I woould suggest to include also this aspect to improve the quality of the review and to get the manuscipt useful for packaging material company.

Response to reviewer comment: Including risk assessment of NIAS into the manuscript would go beyond of scope of this review article. A chapter NIAS is already included and the main paper for NIAS in beverage packaging are given (acetaldehyde, oligomers, inorganic elements from glass).

Round 2

Reviewer 2 Report

The authors have addressed correctly all the comments made by the reviewers.